# Effect of 5-Alpha Reductase Inhibitors on Magnetic Resonance Imaging and Prostate Cancer Detection

**DOI:** 10.3390/biom14020193

**Published:** 2024-02-05

**Authors:** Juan Morote, Natàlia Picola, Jesús Muñoz-Rodriguez, Nahuel Paesano, Xavier Ruiz-Plazas, Marta V. Muñoz-Rivero, Ana Celma, Gemma García-de Manuel, Berta Miró, Pol Servian, José M. Abascal

**Affiliations:** 1Department of Urology, Vall d’Hebron Hospital, 08035 Barcelona, Spain; ana.celma@vallhebron.cat; 2Department of Surgery, Universitat Autònoma de Barcelona, 08193 Bellaterra, Spain; 3Department of Urology, Hospital Universitari de Bellvitge, 08907 Hospitalet de Llobregat, Spain; npicola@bellvitgehospital.cat; 4Department of Urology, Hospital Universitari Parc Taulí, 08208 Sabadell, Spain; jmunoz@tauli.cat; 5Clínica Creu Blanca, 08034 Barcelona, Spain; npaesa@gmail.com; 6Department of Urology, Hospital Universitari Joan XXIII, 43005 Tarragona, Spain; xarupl@gmail.com; 7Department of Urology, Hospital Universitari Arnau de Vilanova, 25198 Lleida, Spain; mviridianam@gmail.com; 8Department of Urology, Hospital Universitari Josep Trueta, 17007 Girona, Spain; gemmagdm@hotmail.com; 9Unit of Statistics and Bioinformatics, Vall d’Hebron Research Institute, 08035 Barcelona, Spain; berta.miro@vhir.org; 10Department of Urology, Hospital Germans Trias i Pujol, 08916 Badalona, Spain; pservian.germanstrias@gencat.cat; 11Department of Urology, Parc de Salut Mar, 08003 Barcelona, Spain; jabascal@psmar.cat

**Keywords:** 5-alpha reductase inhibitors, finasteride, dutasteride, magnetic resonance imaging, prostate cancer, screening, biopsy

## Abstract

Concerns exist regarding the effects of 5-alpha reductase inhibitors (5-ARIs) on multipa-rametric magnetic resonance imaging (mpMRI) and clinically significant prostate cancer (csPCa) detection. Our objective is to analyze the effect of 5-ARI on the prostate imaging–reporting and data system (PI-RADS) distribution and csPCa and insignificant PCa (iPCa) detection. Among 2212 men with serum prostate-specific antigen levels of >3.0 ng/mL and/or suspicious digital rectal examinations who underwent mpMRI and targeted and/or systematic biopsies, 120 individuals exposed to 5-ARI treatment for over a year were identified. CsPCa was defined when the grade group (GG) was >2. The overall csPCa and iPCa detection rates were 44.6% and 18.8%, respectively. Since logistic regression revealed independent predictors of PCa, a randomized matched group of 236 individuals was selected for analysis. The PI-RADS distribution was comparable with 5-ARI exposure (*p* 0.685). The CsPCa detection rates in 5-ARI-naïve men and 5-ARI-exposed men were 52.6% and 47.4%, respectively (*p* 0.596). IPCa was detected in 37.6 and 62.5%, respectively (*p* 0.089). The tumor GG distribution based on 5-ARI exposure was similar (*p* 0.149) to the rates of csPCa and iPCa across the PI-RADS categories. We conclude that exposure to 5-ARI in suspected PCa men did not change the PI-RADS distribution and the csPCa and iPCa detection rates.

## 1. Introduction

The early detection of prostate cancer (PCa) has evolved toward clinically significant PCa (csPCa) [1]. This paradigm shift has resulted from the evidence generated by the European Randomized Study of Screening for Prostate Cancer (ERSPC), which showed a 30% decrease in PCa-specific mortality in the screened population after seven years of follow-up [2]. This decrease has been maintained for 22 years of follow-up in the Göteborg section of the ERSPC [3]. This change was made possible through the introduction and widespread adoption of multiparametric magnetic resonance imaging (mpMRI) for the selection of candidates for prostate biopsy among men suspected of having PCa due to elevated serum prostate-specific antigen (PSA) levels and/or suspicious digital rectal examinations (DREs) [4]. MpMRI can identify lesions with csPCa and classify their risk using the prostate imaging–reporting and data system (PI-RADS) [5]. Today, the role of mpMRI in improving the early detection of csPCa is well accepted [6], although non-negligible insignificant PCa (iPCa) over-detection remains in PI-RADS 3 lesions and systematic biopsies [7].

Benign prostatic hyperplasia (BPH) is a common condition, with over half of men developing symptomatic disease between the ages of 50 and 70 [8], which is the recommended age for the early detection of csPCa [9]. Consequently, many men suspected of having PCa undergo medical treatment for symptomatic BPH. 5-alpha reductase inhibitors (5-ARIs) are one of the primary treatment options, reducing the prostate volume by approximately 20%, alleviating obstructive symptoms, and slowing disease progression. Finasteride and dutasteride (5-ARIs) reduce serum PSA levels by approximately 50% and decrease the risk of developing PCa [10,11,12]. The Prostate Cancer Prevention Trial (PCPT) [11] and the Reduction by Dutasteride of Prostate Cancer Events (REDUCE) trial [12] have shown a reduction in PCa incidence in men receiving finasteride for seven years and dutasteride for four years, respectively, compared with those receiving a placebo, although there was a concerning increase in high-grade Gleason tumors. This initial concern was challenged by the long-term follow-up of the PCPT [13]. The relative increase in high-grade Gleason tumors seems to result from the effective prevention and treatment of low-grade PCa [14]. The role of 5-ARIs in preventing the progression of low-grade PCa under active surveillance is currently under discussion [15].

The MRI in Primary Prostate Cancer after Exposure to Dutasteride (MAPPED) study was designed to provide the radiological effects of six-month exposure to dutasteride on low-grade PCa volume [16]. Preliminary data suggested that dutasteride was associated with an increase in the tumor apparent diffusion coefficient (ADC) and reduced tumor visibility in diffusion-weighted imaging (DWI) without effects on T2 sequences in mpMRI [17,18,19]. Starobinets et al. suggested improved discrimination in mpMRI between the areas with tumors and those with benign tissue in the peripheral zone [20]. However, the effect of 5-ARI exposure on the PI-RADS category and its corresponding detection of csPCa and iPCa in men suspected of having PCa has been poorly analyzed [21,22,23].

Our main goal was to analyze the effect of 5-ARI exposure on the PI-RADS distribution in men suspected of having PCa. Additionally, we aimed to compare the csPCa and insignificant PCa (iPCa) detection rates based on 5-ARI exposure.

## 2. Materials and Methods

### 2.1. Design, Setting, and Participants

This was a retrospective case-control study conducted among 2212 men suspected of having PCa due to serum PSA levels of >3.0 ng/mL and/or suspicious digital rectal examinations (DREs), in whom mpMRI and targeted and/or systematic biopsies were performed at 10 participant centers of the csPCa early detection program in Catalonia during 2022. Catalonia is a Spanish region with 7.9 million inhabitants. A subset of 120 (5.4%) participants were identified as having received 5-ARI treatment for over a year, while 12 individuals were previously excluded from this study due to 5-ARI exposure of less than a year.

### 2.2. MpMRI and Prostate Biopsy Characteristics

MpMRI exams were conducted at each participant center using a pelvic phased-array surface coil and reported with the PI-RADS v.2.1 by experienced radiologists. A magnetic strength field of 1.5 Tesla was utilized in four centers and 3.0 Tesla in six centers. The acquisition protocol followed in all participant centers included T2-weighted imaging (T2W), diffusion-weighted imaging (DWI), and dynamic contrast-enhanced (DCE) imaging, according to the guidelines of the European Society of Urogenital Radiology. MRI-TRUS image fusion was performed for all prostate biopsies using a cognitive technique in five centers and a software technique in five centers. TRUS-assisted prostate biopsies were carried out via the transrectal route in four centers and the transperineal route in six centers. Targeted biopsies, ranging from 2 to 6 cores, were obtained for each suspected lesion (PI-RADS > 3) in addition to a 12-core TRUS systematic prostate biopsy. All included men with negative MRI results (PI-RADS < 3) only underwent a 12-core TRUS systematic biopsy [24]. The prostate biopsies were performed by experienced urologists at each parti-cipant center. The biopsy materials were analyzed in the pathology department of each participant center by experienced uropathologists who used the International Society of Urologic Pathology grade groups (GGs) to classify detected PCa as csPCa when the GG was 2 or higher [25].

### 2.3. Outcome Variables of This Study

The outcome variables of this study were the distribution of PI-RADS categories and the detection rates of csPCa and iPCa.

### 2.4. Statistical Analysis

Statistical analysis was conducted following the harmonization of anonymized datasets. Quantitative variables were defined as medians and interquartile ranges (IQR: 25–75 percentile), while qualitative variables were defined as numbers and percentages. Pearson’s chi-square test was employed to compare the distributions of qualitative variables based on 5-ARI exposure. After univariate analysis, logistic regression was used to identify independent predictive variables for csPCa and iPCa detection. If independent predictive variables for PCa detection were found, a randomized 1:1 matching group was selected based on the binary variable indicating the use of 5-ARI treatment to normalize its effect. Significant differences were considered when the *p*-values were below 0.05. *p*-values between 0.05 and 0.1 were considered as non-significant increasing or decreasing trends. The statistical analysis was conducted using IBM SPSS v.29.0.

## 3. Results

### 3.1. Characteristics of Study Population and Comparison According to 5-ARI Exposure

Table 1 summarizes the overall characteristics of the entire population of men suspected of having PCa and their comparison based on 5-ARI exposure. Notably, 5-ARI u-sers exhibited a significantly higher median age than 5-ARI-naïve individuals (72 vs. 68, respectively; *p* 0.001). The median serum PSA level was similar in both groups (8.0 vs. 7.3 ng/mL; *p* 0.107). A non-significant trend for a higher percentage of suspicious DREs was observed in 5-ARI users (35% vs. 26.8%; *p* 0.057), along with a similar percentage of men with previous negative prostate biopsies (36.7% vs. 31.4%; *p* 0.228) and a family history of PCa (4.2% vs. 8.1%; *p* 0.160). There was a higher median prostate volume in 5-ARI users than in 5-ARI-naïve men (69 vs. 53 cc; *p* < 0.001), accompanied by a lower PSA density (0.12 vs. 0.14; *p* < 0.001).

The distributions of PI-RADS categories according to 5-ARI exposure were similar (*p* 0.238). A non-significant decreasing trend in overall PCa detection was observed in 5-ARI users (55% vs. 63.9%; *p* 0.052), along with similar rates of csPCa detection (42.5% vs. 44.7%; *p* 0.706) and a non-significant decreasing trend in iPCa detection among 5-ARI users (12.5% vs. 19.2%; *p* 0.072).

### 3.2. Search for Independent Predictive Variables of csPCa and iPCa

Considering the significant differences observed in the characteristics between 5-ARI users and 5-ARI-naïve men, we investigated the existence of independent predictive va-riables for csPCa and iPCa. The logistic regression analyses revealed that 5-ARI exposure and a family history of PCa were not independent predictors for overall csPCa and iPCa. Age emerged as an independent predictor for csPCa detection, while serum PSA level was identified as an independent predictive variable for csPCa. Additionally, DRE results, prostate biopsy type (initial vs. repeated), prostate volume, and PI-RADS category were independent predictors of csPCa and iPCa (Table 2).

It was considered essential to normalize the effect of 5-ARI exposure according to the identified independent predictive variables influencing the detection of csPCa and iPCa. A randomized group of 1:1 matched pairs of 5-ARI users and 5-ARI-naïve individuals was selected.

### 3.3. Characteristics of the Randomized Matched Group According to 5-ARI Exposure

Table 3 presents the characteristics of the 118 pairs of men suspected of having PCa, constituting the randomized matched group. Notably, all characteristics were comparable between 5-ARI users and 5-ARI-naïve men, including serum PSA levels (8.0 and 7.5 ng/mL, respectively; *p* 0.304) and the distribution of PI-RADS categories (*p* 0.685), which were not included in the normalization process. The csPCa detection rate was 52.6% in 5-ARI users and 47.4% in 5-ARI-naïve men (*p* 0.596). A non-significant increasing trend in iPCa detection rates was observed in 5-ARI-naïve men compared with 5-ARI users (62.5% vs. 37.5%, respectively; *p* 0.089).

### 3.4. Distribution of csPCa and iPCa in 5-ARI Users and 5-ARI-Naïve Men According to PI-RADS Category

Table 4 illustrates that no significant differences were observed in the detection rates of csPCa and iPCa across the PI-RADS categories. In the subset of men with PI-RADS 4, a non-significant increasing trend in csPCa was found in 5-ARI users compared with 5-ARI-naïve men (46.4% vs. 28.6%; *p* 0.078). Additionally, a non-significant decreasing trend in iPCa detection was noted in men with PI-RADS 5 in 5-ARI users compared with 5-ARI-naïve men (3.7% vs. 16.7%; *p* 0.080).

### 3.5. Distribution of Grade Groups in Tumors Detected in 5-ARI Users and 5-ARI-Naïve Men

The distributions of the GGs of tumors detected were comparable between 5-ARI u-sers and 5-ARI-naïve men (*p* 0.149), as illustrated in Figure 1. High-grade PCa (GG > 3) was observed in 57% of tumors detected in 5-ARI users compared with 38.6% in 5-ARI-naïve men (*p* 0.146).

## 4. Discussion

The present study reported unexpected comparable serum PSA levels in 5-ARI users and 5-ARI-naïve men suspected of having PCa. We note that reported serum PSA levels in 5-ARI users corresponded to the real measurements without any adjustment. This observation was true for the entire population and the selected matched group for analysis. Similar findings have been reported by Kim et al. [21] and Wang et al. [23]. This contrasts with the expected lower serum PSA levels in 5-ARI users compared with those observed in 5-ARI-naïve men because of the effect of 5-ARI exposure on serum PSA levels. This observation may be due to not following the recommendation to closely monitor serum PSA levels after reaching their nadir and to establish suspicion of PCa based on a confirmed increase in levels higher than 0.3 ng/mL [26]. Chang et al. recently underlined the necessity to closely monitor serum PSA levels in patients exposed to 5-ARI treatment to prevent delays in diagnosing high-grade PCa [27].

This is the first study analyzing the effect of 5-ARI exposure in a randomized matched group to normalize the influence of confusing independent variables for csPCa and iPCa. The present study suggests that 5-ARI exposure does not modify the distribution of PI-RADS categories. Furthermore, we observed comparable detection rates of csPCa and iPCa according to 5-ARI exposure. However, a non-significant decreasing trend in iPCa detection was noted in men exposed to 5-ARI. This non-significant decreased trend was observed in both the overall population and the randomized matched group. Kim et al. first reported the effects of 5-ARI exposure on the PI-RADS distribution and the detection of csPCa in 2019 [21]. Among 706 men with suspected PCa undergoing targeted and systematic biopsies, 80 (11.3%) were identified as receiving 5-ARI treatment for over a year. The authors found a similar distribution of PI-RADS categories between 5-ARI users and 5-ARI-naïve men. Additionally, comparable rates of csPCa and iPCa were observed. In 2023, Wang et al. reported a study conducted on 351 individuals suspected of having PCa who underwent saturation biopsies and targeted biopsies. They identified 54 (15.3%) individuals undergoing 5-ARI treatment for over a year. The authors observed a comparable distribution of PI-RADS categories between 5-ARI users and 5-ARI-naïve men. However, they found a significant reduction in overall PCa detection in 5-ARI users compared with 5-ARI-naïve men (68.0% vs. 46.3%, respectively), with a similar rate of csPCa. Although data on iPCa were not reported, a decreased rate of iPCa in 5-ARI users can be inferred in this series. The findings from this study align with those of previously mentioned studies, even though these analyses were not conducted in randomized matched groups.

The recently reported Prostate MRI Outcome Database (PROMOD) study included 705 men receiving 5-ARI treatment for more than three months and 6913 5-ARI-naïve men. The study involved mpMRI and targeted and/or systematic biopsies performed at 36 centers between 2020 and 2022. The authors concluded that 5-ARI exposure did not affect the PI-RADS distribution and its association with csPCa detection. The rate of csPCa was comparable in both groups, although a higher rate of high-grade PCa (GG > 3) was observed in 5-ARI users with PI-RADS 5. The median serum PSA levels reported in this study were 6.0 ng/mL in 5-ARI users and 6.5 ng/mL in 5-ARI-naïve men. Although a significant difference was reported between both serum PSA levels, those in 5-ARI users were notably higher than those expected [28]. In 2021, Forte et al. evaluated the PI-RADS v.2.0 in 75 men suspected of having PCa who underwent 5-ARI treatment. They concluded that the PI-RADS v.2.0 exhibited good accuracy in predicting csPCa [22]. In 2021, Artiles et al. concluded that 5-ARI exposure was an independent predictor of csPCa after analyzing 34 men suspected of having PCa with negative mpMRI results who underwent saturation biopsies [29].

The limitations of the present study are its retrospective design and the small size of the case group. The non-significant trends observed may be significant differences with an appropriate size previously calculated. Another limitation is that the length of 5-ARI exposure was not controlled beyond one year, possibly not allowing adequate time to observe changes in the PI-RADS distribution. Notably, the PCPT had a follow-up period of seven years, while the REDUCE trial spanned four years [11,13]. Additionally, the csPCa definition used in prostate biopsies may not fully represent the true pathology in entire prostate specimens.

In summary, few studies have analyzed the effect of 5-ARI exposure on the PI-RADS distribution and the corresponding detection rates of csPCa and iPCa. There is no evidence of changes in PI-RADS categories secondary to 5-ARI exposure. While there is a recognized delay in prostate biopsies for men undergoing 5-ARI based on serum PSA levels, our findings indicate no significant changes in the PI-RADS distribution and no significant differences in csPCa and iPCa detection rates related to 5-ARI exposure.

## 5. Conclusions

The PI-RADS score distribution in 5-ARI users was similar to that in 5-ARI-naïve men. The detection rates for csPCa and iPCa were also similar in both groups of men suspected of having PCa.

## Figures and Tables

**Figure 1 biomolecules-14-00193-f001:**
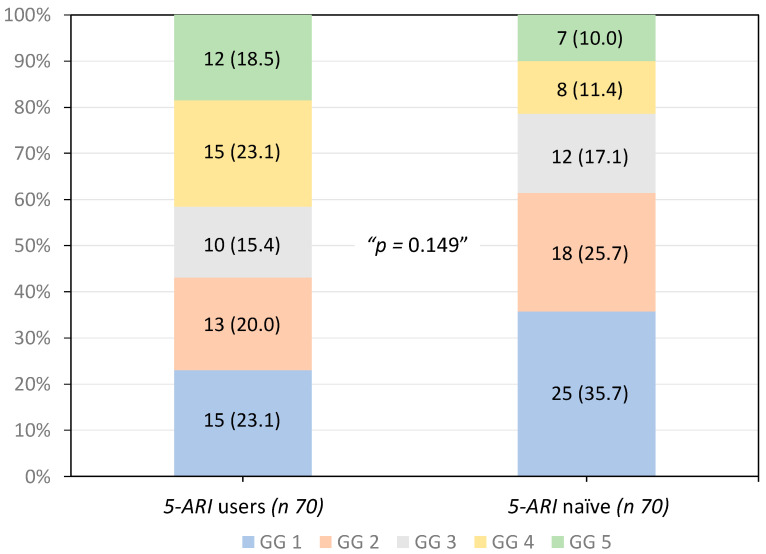
Distribution of the ISUP grade groups of tumors detected according to the 5-ARI exposure.

**Table 1 biomolecules-14-00193-t001:** Characteristics of included population according to the 5-ARI exposure.

Characteristic	5-ARI Users	5-ARI Naïve	*p* Value
Number of men, (%)	120 (5.4)	2092 (94.6)	-
Median age (IQR), years	72 (68–76)	68 (62–73)	<0.001
Median PSA (IQR), ng/mL	8.0 (5.0–13.0)	7.3 (5.3–11.0)	0.107
Suspicious DRE, *n* (%)	42 (35.0)	561 (26.8)	0.057
Repeated biopsy, *n* (%)	44 (36.7)	657 (31.4)	0.228
PCa family history, (%)	5 (4.2)	169 (8.1)	0.160
Median prostate volume, cc (IQR), cc	69 (49–97)	53 (38–74)	<0.001
PSA density, ng/mL/cc	0.12 (0.08–0.21)	0.14 (0.09–0.22)	0.025
PI-RADS score, *n* (%)			
≤2	15 (12.5)	320 (14.5)	0.328
3	20 (16.7)	429 (20.5)	
4	57 (47.5)	903 (43.2)	
5	28 (23.6)	440 (21.0)	
PCa, *n* (%)	66 (55.0)	1336 (63.9)	0.052
csPCa, *n* (%)	51 (42.5)	935 (44.7)	0.706
iPCa, *n* (%)	15 (12.5)	401 (19.2)	0.072

5-ARI, 5-alpha reductase inhibitor; IQR, interquartile range; PSA, prostate-specific antigen; DRE, digital rectal examination; PCa, prostate cancer; PI-RADS, Prostate imaging report and data system; csPCa, clinically significant PCa; iPCa, insignificant PCa.

**Table 2 biomolecules-14-00193-t002:** Multivariate analysis to identify independent predictive variables for csPCa and iPCa detection.

Predictive Variable	csPCa		iPCa	
	Odd Ratio (95% CI)	*p* Value	Odd Ratio (95% CI)	*p* Value
5-ARI exposure, Ref. no	0.826 (0.521–1.311)	0.418	0.628 (0.357–1.104)	0.106
Age, Ref. year	1.059 (1.044–1.072)	<0.001	1.010 (0.995–1.024)	0.186
Serum PSA, Ref. ng/mL	1.014 (1.006–1.021)	<0.001	0.986 (0.976–0.997)	0.012
DRE. Ref. normal	2.254 (1.781–2.853)	<0.001	0.626 (0.475–0.825)	<0.001
Type of biopsy, Ref. initial	0.792 (0.633–0.990)	0.040	1.780 (1.420–2.230)	<0.001
PCa family history, Ref. no	1.230 (0.853–1.772)	0.267	1.166 (0.793–1.713)	0.435
Prostate volume, Ref. cc	0.980 (0.976–0.983)	0.001	0.997 (0.993–1.001)	0.092
PI-RADS score, Ref. ≤ 2	2.350 (2.277–2.855)	0.001	0.907 (0.832–0.989)	0.027

5-ARI, 5-alpha reductase inhibitor; PSA, prostate-specific antigen; DRE, digital rectal examination; PCa, prostate cancer; PI-RADS, Prostate imaging report and data system; csPCa, clinically significant PCa; iPCa insignificant PCa.

**Table 3 biomolecules-14-00193-t003:** Characteristics of randomized matched group analyzed based on exposure to 5-ARI.

Characteristic	5-ARI Users	5-ARI Naïve	*p* Value
Number of men, (%)	118 (50.0)	118 (50.0)	-
Median age (IQR), years	72 (68–76)	71 (68–76)	0.799
Median PSA (IQR), ng/mL	8.0 (5.2–12.9)	7.5 (5.6–10.6)	0.304
Suspicious DRE, *n* (%)	42 (35.6)	43 (36.4)	0.892
Repeated biopsy, *n* (%)	43 (36.4)	43 (36.4)	1.000
PCa family history, (%)	5 (4.2)	5 (4.2)	1.000
Median prostate volume (IQR), cc	67 (49–95)	66 (49–96)	0.852
PI-RADS score, *n* (%)			
≤2	15 (45.5)	18 (54.5)	0.685
3	20 (58.8)	14 (41.2)	
4	56 (50.0)	56 (50.0)	
5	27 (47.4)	30 (52.6)	
PCa, *n* (%)	65 (48.1)	70 (51.9)	0.599
csPCa, *n* (%)	50 (52.6)	45 (47.4)	0.596
iPCa, *n* (%)	15 (37.5)	25 (62.5)	0.089

5-ARI, 5-alpha reductase inhibitor; IQR, interquartile range; PSA, prostate-specific antigen; DRE, digital rectal examination; PCa, prostate cancer; PI-RADS, prostate imaging report and data system; csPCa, clinically significant PCa; iPCa, insignificant PCa.

**Table 4 biomolecules-14-00193-t004:** Distribution of csPCa and iPCa according to the 5-ARI exposure and the PI-RADS score.

PI-RADS Score	*n*	csPCa			iPCa		
		5-ARI Users	5-ARI Naïve	*p* Value	5-ARI Users	5-ARI Naïve	*p* Value
≤3, *n* (%)	33	0/15 (0)	3/18 (16.7)	0.233	1/15 (6.7)	1/18 (5.6)	0.894
3, *n* (%)	34	2/20 (10.0)	3/14 (21.4)	0.627	3/20 (15.0)	1/14 (7.1)	0.627
4, *n* (%)	112	26/56 (46.4)	16/56 (28.6)	0.078	10/56 (17.9)	18/56 (32.1)	0.126
5, *n* (%)	57	22/27 (81.5)	23/30 (76.7)	0.751	1/27 (3.7)	5/30 (16.7)	0.080

csPCa, clinically significant prostate cancer; iPCa, insignificant prostate cancer; PI-RADS, prostate imaging report and data system; 5-ARI, 5-alpha reductase inhibitor.

## Data Availability

The data presented in this study are available on request from the corresponding author.

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
