# Peer review of "Effect of 5-Alpha Reductase Inhibitors on Magnetic Resonance Imaging and Prostate Cancer Detection"

_biomolecules, 2024, doi:10.3390/biom14020193_

Round 1

Reviewer 1 Report

Comments and Suggestions for Authors

The article titled "Effect of 5-Alpha Reductase Inhibitors on Magnetic Resonance Imaging and Prostate Cancer Detection" presents a study on the impact of 5-alpha reductase inhibitors (5-ARIs) on multiparametric magnetic resonance imaging (mpMRI) in prostate cancer detection. The authors aim to analyze the effect of 5-ARI on the prostate imaging–reporting and data system (PI-RADS) distribution and the detection of clinically significant prostate cancer (csPCa) and insignificant PCa (iPCa).

The study is a retrospective case–control investigation. It involves 2212 men suspected of prostate cancer due to elevated serum prostate-specific antigen (PSA) levels and/or suspicious digital rectal examinations. The study compares csPCa and iPCa detection rates based on 5-ARI exposure. It utilizes mpMRI and targeted/systematic biopsies for data collection. The retrospective design, while valid, may introduce biases associated with historical data, such as inconsistencies in mpMRI protocols across different centers. The article could benefit from a more detailed description of the mpMRI protocol, considering variations in MRI machines and settings (1.5 Tesla vs. 3.0 Tesla). 

The study's critical weaknesses primarily revolve around its retrospective design, which inherently limits control over variables and introduces potential biases. A significant concern is the homogeneity of the participant groups, particularly regarding variables like age and prostate volume, which could influence the outcomes but were not thoroughly analyzed. Additionally, the study could have benefitted from a more detailed description of the mpMRI protocols used, considering the potential variability in imaging quality and interpretation across different centers. The lack of a multicenter approach and standardized mpMRI protocols may also limit the generalizability of the findings. Furthermore, the study does not extensively compare mpMRI with traditional diagnostic methods, missing an opportunity to delineate the specific advantages or challenges of mpMRI in the context of 5-ARI treatment. These areas present opportunities for future research to build upon and address the limitations noted.

Comments on the Quality of English Language

Minor language editing required 

Author Response

Response to Comments and Suggestions for Authors. Reviewer 1#

The article titled "Effect of 5-Alpha Reductase Inhibitors on Magnetic Resonance Imaging and Prostate Cancer Detection" presents a study on the impact of 5-alpha reductase inhibitors (5-ARIs) on multiparametric magnetic resonance imaging (mpMRI) in prostate cancer detection. The authors aim to analyze the effect of 5-ARI on the prostate imaging–reporting and data system (PI-RADS) distribution and the detection of clinically significant prostate cancer (csPCa) and insignificant PCa (iPCa).

The study is a retrospective case–control investigation. It involves 2212 men suspected of prostate cancer due to elevated serum prostate-specific antigen (PSA) levels and/or suspicious digital rectal examinations. The study compares csPCa and iPCa detection rates based on 5-ARI exposure. It utilizes mpMRI and targeted/systematic biopsies for data collection. The retrospective design, while valid, may introduce biases associated with historical data, such as inconsistencies in mpMRI protocols across different centers. The article could benefit from a more detailed description of the mpMRI protocol, considering variations in MRI machines and settings (1.5 Tesla vs. 3.0 Tesla). 

The study's critical weaknesses primarily revolve around its retrospective design, which inherently limits control over variables and introduces potential biases. A significant concern is the homogeneity of the participant groups, particularly regarding variables like age and prostate volume, which could influence the outcomes but were not thoroughly analyzed. Additionally, the study could have benefitted from a more detailed description of the mpMRI protocols used, considering the potential variability in imaging quality and interpretation across different centers. The lack of a multicenter approach and standardized mpMRI protocols may also limit the generalizability of the findings. Furthermore, the study does not extensively compare mpMRI with traditional diagnostic methods, missing an opportunity to delineate the specific advantages or challenges of mpMRI in the context of 5-ARI treatment. These areas present opportunities for future research to build upon and address the limitations noted.

Thank you for your comments on the study.

Q.1. The article could benefit from a more detailed description of the mpMRI protocol, considering variations in MRI machines and settings (1.5 Tesla vs. 3.0 Tesla). 

A.1. We agree with your comment and suggestion. We have introduced the next paragraph in lines 97-100: “The acquisition protocol followed in all participant centers included T2-weighted imaging (T2W), diffusion-weighted imaging (DWI) and dynamic contrast-enhanced (DCE) imaging, according to guidelines of the European Society of Urogenital Radiology.”

Q.2.  Minor language editing required.

A.2.  A new proof reading of English has been performed. 

Reviewer 2 Report

Comments and Suggestions for Authors

This is a retrospective analysis of a series of patients submitted to PCa screening and eventual MpMRI and biopsy. The aim of the study is to find out the impact of 5ARI use on PIRADS distribution. Authors concluded that there is no difference between 5 ARI users and non-users. The paper is well written, clear, and concise. 

There are some minor issues. 

1) How did Authors calculate baseline PSA for 5ARI users? Median PSA is similar in the two groups of patients. Was it the PSA effectively measured or was it doubled? The same issue should be clarified regarding PSA density. The discussion does not help to understand.

2) It seems that the incidence of PCa and of iPCA was "clinically" significantly higher in the non-5ARI population. It may be deducted the statistical significance would be reached with a greater population (respectively P 0.052, P 0.07) as Authors correctly suggest in the discussion. Indeed, these results are similar to previous findings and confirm the validity of the study even if retrospective that is the main limitation of this nice study. It should be stated in the discussion.

Author Response

Response to Comments and Suggestions for Authors. Reviewer 2#

This is a retrospective analysis of a series of patients submitted to PCa screening and eventual MpMRI and biopsy. The aim of the study is to find out the impact of 5ARI use on PIRADS distribution. Authors concluded that there is no difference between 5 ARI users and non-users. The paper is well written, clear, and concise. 

There are some minor issues. 

Q.1.  How did Authors calculate baseline PSA for 5ARI users? Median PSA is similar in the two groups of patients. Was it the PSA effectively measured or was it doubled? The same issue should be clarified regarding PSA density. The discussion does not help to understand.

A.1.  The serum PSA levels in 5-ARI users reported as measured before the mpMRI and biopsy corresponded to the real PSA, it was not modified. That´s the reason of the first comment in the discussion, lines 210-220. To clarify this setting, we added in lines 211-212 the sentence: “We note that reported serum PSA levels in 5-ARI users corresponded to the real measurements without any adjustment”

Q.2.  It seems that the incidence of PCa and of iPCa was "clinically" significantly higher in the non-5ARI population. It may be deducted the statistical significance would be reached with a greater population (respectively P 0.052, P 0.07) as Authors correctly suggest in the discussion. Indeed, these results are similar to previous findings and confirm the validity of the study even if retrospective that is the main limitation of this nice study. It should be stated in the discussion.

  1. 2. We agree with this comment, and we changed the first sentence in the limitations paragraph according to the suggestion in lines 259-261: “Limitations of the present study are it´s retrospective design and the small size of the case group. The non-significant trends observed may be significant differences with an appropriate size previously calculated.”